# Public Discourse Surrounding Suicide during the COVID-19 Pandemic: An Unsupervised Machine Learning Analysis of Twitter Posts over a One-Year Period

**DOI:** 10.3390/ijerph192113834

**Published:** 2022-10-24

**Authors:** Shu Rong Lim, Qin Xiang Ng, Xiaohui Xin, Yu Liang Lim, Evelyn Swee Kim Boon, Tau Ming Liew

**Affiliations:** 1Health Services Research Unit, Singapore General Hospital, Singapore 169608, Singapore; 2MOH Holdings Pte Ltd., 1 Maritime Square, Singapore 099253, Singapore; 3Department of Psychology, Singapore General Hospital, Singapore 169608, Singapore; 4Department of Psychiatry, Singapore General Hospital, Singapore 169608, Singapore; 5SingHealth Duke-NUS Medicine Academic Clinical Programme, Duke-NUS Medical School, Singapore 169857, Singapore; 6Saw Swee Hock School of Public Health, National University of Singapore, Singapore 117549, Singapore

**Keywords:** suicide, COVID-19, strain theory, machine learning, topic modelling, social media

## Abstract

Many studies have forewarned the profound emotional and psychosocial impact of the protracted COVID-19 pandemic. This study thus aimed to examine how individuals relate to suicide amid the COVID-19 pandemic from a global perspective via the public Twitter discourse around suicide and COVID-19. Original Twitter tweets from 1 February 2020 to 10 February 2021 were searched, with terms related to “COVID-19”, “suicide”, or “self-harm”. An unsupervised machine learning approach and topic modelling were used to identify topics from unique tweets, with each topic further grouped into themes using manually conducted thematic analysis by the study investigators. A total of 35,904 tweets related to suicide and COVID-19 were processed into 42 topics and six themes. The main themes were: (1) mixed reactions to COVID-19 public health policies and their presumed impact on suicide; (2) biopsychosocial impact of COVID-19 pandemic on suicide and self-harm; (3) comparing mortality rates of COVID-19, suicide, and other leading causes of death; (4) mental health support for individuals at risk of suicide; (5) reported cases and public reactions to news related to COVID-19, suicide, and homicide; and (6) figurative usage of the word suicide. The general public was generally concerned about governments’ responses as well as the perturbing effects on mental health, suicide, the economy, and at-risk populations.

## 1. Introduction

Worldwide, many countries enacted lockdowns and social and movement restrictions due to the coronavirus disease 2019 (COVID-19) in a bid to flatten the curve of COVID-19 infections. Although public health interventions, such as physical distancing and self-isolation, do significantly reduce the transmission of COVID-19, these restrictions have unintended implications in multiple facets of human life [1], from rising unemployment rates and fears of impending economic turmoil [2,3] to negative mental health repercussions associated with COVID-19 (e.g., elevated levels of anxiety and depressive symptoms [4]. Many studies have warned of profound mental health impacts as well as the escalation of suicide risks given the emotional and psychosocial effects of anticipated economic downturns and nation-wide lockdowns, with suicide as one of the psychological sequelae of the COVID-19 pandemic [5,6].

The number of deaths by suicide on a global scale is approximately 800,000 annually; for every completed adult suicide, there could have been more than 20 suicide attempts [7]. Predictive models estimated an increase of approximately 2135 to 9570 suicides per year in association with projected figures of 5.3 to 24.7 million global job losses due to COVID-19, respectively [3], equivalent to an approximate range of 0.26% to 1.2% increase in suicide worldwide.

One theoretical perspective from suicide literature that can aid in the understanding and prevention of suicide during the COVID-19 pandemic is the strain theory of suicide (STS) [8]. The STS theorized that suicide is preceded by the experience of psychological strain from at least two competing pressures that are at constant odds with each other [8]. According to STS, suicide may be triggered by psychological strains related to: (a) conflicting social values or beliefs; (b) aspiration and reality incongruence; (c) relative deprivation, i.e., when one perceives others to be comparatively better; and (d) deficient coping, i.e., inability to cope with life crisis [8,9]. When one of these strains becomes unbearable, individuals may resort to inward violence, i.e., suicide, as a radical solution [8]. Various studies by Zhang et al. found evidence in support of STS in rural China [10] and the southwestern United States [9], albeit these studies were from non-pandemic periods. Two recent systematic reviews [11,12] identified some evidence linking prior public health crises and suicide, suggesting an increase in suicide risks and rates during disease outbreaks, e.g., the 1889 Russian influenza, 1918 Great Influenza, 2003 severe acute respiratory syndrome, and 2014 Ebola outbreaks. Compared to previous epidemics and pandemics, however, the current COVID-19 pandemic has resulted in significant and widespread economic and societal disruptions across the globe.

At the time of writing, the pandemic has been ongoing for more than two years, with much mental and physical fatigue felt by the general population [13]. Given the growing mental health concerns related to COVID-19 and suicide, and to aid in suicide prevention during this protracted pandemic, the present study sought to examine how individuals relate to suicide amid the COVID-19 crisis from a global perspective—specifically, the public Twitter discourse that revolves around suicide and COVID-19.

## 2. Methods

In this study, data from the Twitter platform were analyzed using an unsupervised machine learning approach of topic modelling [14]. The selection of Twitter as the social site for data collection is due to its accessibility as a large volume of global data. Furthermore, Twitter is an ideal microblogging social media site for analysis of naturally occurring data to examine how suicide is talked about during COVID-19 pandemic, as users can upload messages of up to 280 characters of text, also referred to as “tweets”. Previous studies have found natural language processing techniques and such social media analyses to be a feasible and novel method to study public sentiment and emotional manifestations occurred on a given topic [15,16,17,18].

Original tweets, i.e., not retweets or duplicate tweets, posted in English between 1 February 2020 and 10 February 2021 (both dates inclusive) were searched in a publicly available COVID-19 Twitter dataset [19]. The search terms were ‘COVID-19′, and “suicide” or “self-harm”. Spelling variants: “COVID”, “COVID_19”, “COVID 19”, “COVID19”, “coronavirus”, “corona virus”, “2019ncov”, “coronavirusPandemic”, “CoronaOutbreak”, “WuhanVirus”, “suicidal”, “suicidality”, “selfharm”, and “self harm” were also included. Only tweets on suicide by individual users were included, while tweets by news media agencies and organizations were excluded. Named entity recognition, a natural language processing machine learning approach, was used to process human name recognition [20].

Next, during the pre-processing phase, sentences from each tweet were tokenized (i.e., reduced to single words), and cleaned (i.e., punctuation and extra spaces were removed, and words were converted into lowercase [14,17]. English words with high frequencies of occurrence that add little to the meaning of sentences, also known as stop words, were removed (e.g., “and”, “the”, “a”, “of”). Lemmatization was performed to transform words to their base forms (e.g., “best” is transformed to “good”, “friends” to “friend”). Subsequently, unsupervised bag-of-words text analysis, specifically the topic modelling technique, was applied to identify clusters of words that tend to co-occur. High probability co-occurring words are considered to be of the same “topic” as per previous studies [17]. Topic modelling allows a large number of texts to be analyzed using machine learning algorithms, thereby reducing the amount of time taken to analyze data manually. To enhance accuracy of the model in identifying topics, three covariates were added to the model: continent location of tweets, date of tweets, and number of followers. Next, to identify the optimal number of topics, Mimno and Lee’s (2014) algorithm was applied [21].

Output from topic modelling was examined by three study investigators (S.R.L., Q.X.N. and T.M.L.) to ensure coherence of the identified topics. Descriptive labels for the topics were manually crafted by the same study investigators based on identified keywords and sample tweets of each topic. Lastly, thematic analysis was performed through an inductive and iterative process as introduced by Braun and Clarke (2006), with (1) study investigators (S.R.L., Q.X.N., X.X., Y.L.L., E.S.K.B., and T.M.L) familiarizing themselves with the keywords and sample tweets; (2) producing preliminary codes; (3) formulating overarching themes; (4) reviewing and refining themes; (5) defining and specifying themes; and (6) producing a write-up [22]. Thematic analysis was preferred as it allowed flexibility and can be applied across a range of theoretical and epistemological approaches. Inter-coder reliability was not calculated as the authors reviewed the codes and themes independently; coding disagreements between authors were resolved by further discussion until consensus was reached.

R (version 3.6.3) was used for all analyses. For data processing, the following packages were used: “spacyr” (version 1.2.1) for identifying individual users’ accounts, “quanteda” for text pre-processing, and “stm” for topic modelling. Data extracted for this study were in accordance with Twitter’s terms.

## 3. Results

A total of 109,014 initial tweets were identified in the period of 1 February 2020 to 10 February 2021. After removing duplicate tweets, tweets by organizations, and tweets without the relevant terms of “COVID-19”, “suicide”, or “self-harm”, 35,904 tweets remained. A flowchart on tweet selection is shown in Figure 1. Geographical distribution of tweets is presented in Figure 2.

The majority of tweets with location information were from North America (25.9%) and Europe (11.5%), with the remainder from Asia (5.2%), Australia (2.6%), Africa (1.6%), South America (0.6%), and unknown location (52.7%).

The current analysis generated 42 topics related to suicide and the COVID-19 pandemic; they were clustered into six themes. Figure 3 shows the word clouds for the six themes; Table 1 contains further details of the topics within each theme.

### 3.1. Theme 1: Mixed Reactions to COVID-19 Public Health Policies and Their Presumed Impact on Suicide

Theme 1 comprised of 11 topics, accounting for 25.7% of tweets, depicting the diverse responses toward COVID-19 public health policies. For example, Topic 4: Skepticisms about the governments’ COVID-19 pandemic management; Topic 9: Inculpating the politicians for job losses and suicide death tolls. Additionally, there were contrasting public responses toward attending school and participating in exams during the pandemic, illustrated by Topic 29: Students’ protests over taking exams during COVID-19 pandemic versus Topic 5: Advocating for reopening of schools during COVID-19 lockdowns.

### 3.2. Theme 2: Biopsychosocial Impact of COVID-19 Pandemic on Suicide and Self-Harm

This theme composed 20.9% of tweets and included 10 topics. The associated biopsychosocial impacts of COVID-19 pandemic alluded to within Theme 2 are higher rates of mental health issues, suicide and mortality, drug and substance abuse, domestic abuse, divorce, unemployment, poverty, and violence (e.g., Topic 35: Associating the exacerbation of domestic abuse, drug abuse, and gun violence to COVID-19 pandemic; Topic 1: Missed diagnosis and treatment delay for other medical conditions).

### 3.3. Theme 3: Comparing Mortality Rates of COVID-19, Suicide, and Other Leading Causes of Death

Theme 3 contained six topics, accounting for 19.3% of tweets. An instance of the leading causes of deaths comparisons is depicted by Topic 28: Mortality rates of COVID-19 against other leading causes of death by presenting the various diseases mortality data (e.g., COVID-19, heart disease, accidents, and stroke).

### 3.4. Theme 4: Mental Health Support for Individuals at Risk of Suicide

There were seven topics within Theme 4, accounting for 17.0% of tweets. This theme comprises of topics on mental health awareness, mental health crisis, and increased suicide risks for certain vulnerable populations (e.g., Topic 23: Mental health crisis experienced by healthcare workers during COVID-19 pandemic; Topic 25: Increased suicide risks of military veterans), and advocating for mental health support (i.e., Topic 38: Mental health support for suicide at-risk loved ones).

### 3.5. Theme 5: Reported Cases and Public Reactions to News Related to COVID-19, Suicide, and Homicide

Theme 5 comprised six topics with a combined prevalence rate of 13.6%. Several high-profile COVID-19 related suicide and homicide form distinct topics. For instance, Topic 21: Death of a finance minister over the economic state from COVID-19; Topic 14: Suicide death of an emergency room doctor; and Topic 39: News reports of the death of a COVID-19 researcher who was close to making significant findings. In combination, these three topics accounted for 6.9% of tweets. Distinctively within Theme 5, Topic 40: Combatting the spread of COVID-19 and self-harm misinformation and disinformation had the highest prevalence rate, at 2.6%.

### 3.6. Theme 6: Figurative Usage of the Word Suicide

The last theme, comprising only two topics, showed the word suicide being used as a figure of speech. Specifically, the COVID-19 vaccine, and refusal for mask wearing and social distancing were metaphorized as suicide, accounting for 3.6% of tweets. The name “Robert Sherriff” was the most common word in the word cloud, who may have multiple Twitter accounts (poet, singer, author, historian etc.). The other words “sing”, “song” and “robertsheriff871954” could have been referring to his COVID-19 song, which expressed pessimism that, “COVID-19 has got me crying… I got so many friends out there just dying.”

Notably, some of these words in the word cloud (Figure 3) were not identified in Table 1. Reasons for the discrepancy between the Table 1 keywords and the Theme 6 word cloud could be that term frequency–inverse document frequency (TF-IDF), which was used to generate the word clouds, may give more emphasis to words that are unique to each theme, and larger items in the word cloud indicate more informative feature (higher TF-IDF scoring) [23].

## 4. Discussion

This study examined the suicide and COVID-19 discourse on social media during COVID-19 pandemic from 1 February 2020 to 10 February 2021. Using unsupervised machine learning, 35,904 tweets relevant to suicide and COVID-19 were distilled to 42 topics and thematically categorized into six themes.

There was a diverse range of conversations, with underlying tensions, conundrums, and mixed reactions in public response to governments’ COVID-19 public health policies and their presumed impact on suicide. This is to be expected and was clearly reflected in the dominant Theme 1. Majority of topics within Theme 1 were criticisms towards the governments and politicians’ COVID-19 response. It can be inferred that the general public was dissatisfied with COVID-19 measures as the public discourse within Theme 1′s more prevalent topics spoke of unemployment, economic suicide, and more deaths from lockdowns and related measures than from the pandemic itself. An earlier study found that although discussion topics on Twitter around COVID-19 did change over time, negative sentiments persisted, and people had a generally negative outlook towards COVID-19 [24]. The mixed responses may be in part due to countries and cities imposing different levels of closures and stringency on restrictions to human movement and COVID-19 preventive measures [25]. This potentially reflects the levels of trust the public has in governments [26], particularly in times of uncertainty as conditions of the pandemic evolve each day. Elevated mental distress during pandemic lockdowns [4,27], feelings of uncertainty, and lack of control or mastery over fate are risks factors of suicide [28].

The next dominant theme is Theme 2: Biopsychosocial impact of COVID-19 pandemic on suicide and self-harm. The discourse within Theme 2 revealed feelings of general hopelessness, despair, and desperation. These are in relation to the pandemic policies’ impact on suicide, mental and physical health, and the human social conditions. Previous studies have also highlighted the unintended negative impact of the global COVID-19 pandemic on mental health and suicide risks [4,28,29].

The general sentiments of Theme 3 were aptly demonstrated by Topic 28: Mortality rates of COVID-19 against other leading causes of death. Here, the mortality rates of COVID-19 are positioned against other leading causes of death (e.g., heart disease, cancer, and stroke) for comparison. These comparisons are made to achieve certain social functions; for instance, to raise awareness and a call-to-action for the high death rates of specific leading causes of death. Alternatively, it can be argued that it serves to deflect attention on the high COVID-19 mortality rates by comparing COVID-19 to other more life-threatening diseases (e.g., heart disease and cancer) to justify and strengthen the argument for re-opening countries.

Although most topics in Theme 3 were about comparing and contrasting the leading causes of death during the COVID-19 pandemic, one topic that potentially instilled some optimism during the lockdown isolating period was Topic 10: Positive effect of lockdown on suicide rates in Japan. In Japan, suicide trends decreased during the beginning of the COVID-19 wave, from February to June 2020 [6]. Such a trend was also observed in other countries, such as Norway [30] and Peru [31]. However, during Japan’s second COVID-19 wave, suicide rates increased from July to October 2020, particularly for female adults, children, and teens [4]. The initial decline in suicide rates during the first wave is transient; this phenomenon is also referred to as the “honeymoon effect” [32] or the pulling-together effect [33]. According to natural disasters and suicide literature, such phenomenon could be attributed to the facilitation of having a collective purpose in society as individuals work together on bringing down COVID-19 infection rates, and when in the presence of an actual life-threatening risk (in this instance, COVID-19), individuals cherish living more than dying [32,33,34]. Hence, the initial decline in suicide rates.

Theme 4: Mental health support for individuals at risk of suicide during COVID-19 pandemic highlighted a pertinent social concern, where many patients faced limited access to psychiatric services and social isolation during the pandemic [35]. Topic 24: Advocating for better mental health support highlighted individuals rallying others on twitter during such a trying period. Individuals rallied at-risk individuals to speak openly to loved ones about their mental health concerns and suicidal ideation. Indeed, having open conversations about mental health and suicide may help to reduce social stigma, raise public awareness, and encourage help-seeking behaviors for at-risk individuals [36].

Theme 5 pertained to reported cases and public reactions to news related to COVID-19, suicide, and homicide. Several high-profile COVID-19 related suicide and homicide cases in the news entered the Twitter public discourse, forming the larger proportion of topics within Theme 5, e.g., “German minister commits suicide after ‘coronavirus crisis worries’.” While it is premature to draw conclusions on the causes of suicidal behavior during COVID-19 based on news reports, this seems to be aligned with STS-deficient coping as one of the apparent sources of psychological strain experienced by individuals who reportedly died by suicide [8]. During the pandemic, news consumption increased as well, as the public relied more on the internet for timely news updates and to stay informed about public health measures on the pandemic [37]. This was demonstrated by the various topics within Theme 5, as the Twitter discourse was about COVID-19 related news, homicide and suicide cases of unrelated individuals. High social media consumption can be a doubled-edged sword: while individuals may be able to stay abreast of the latest health advisories, higher social media consumption can also result in information overload and exacerbate feelings of anxiety and distress [38,39], which can be contagious even via social media platforms [40]. Additionally, social media platforms are rife with misinformation and disinformation [41], and they can undermine effective public health communication and result in further distress. This was highlighted by Topic 40: Combatting the spread of COVID-19 and self-harm misinformation and disinformation.

For theme 6, COVID-19 vaccination, refusal for mask wearing and social distancing were metaphorized, i.e., framed figuratively by drawing on the suicide metaphor. Metaphor, a form of figure of speech, is a commonly used cognitive and communication device [42]. Metaphor allows the topic to provoke thought, make clarifications, and to construct emphatic similarities comparisons [43]. Additionally, metaphor allows the framed topic to be made more salient for problem definition, causes identification, moral judgment, and action suggestion [44]. As demonstrated in Topic 33, COVID-19 vaccination is presented as mortally harmful so as to influence the decision to not vaccinate against COVID-19, whereas in Topic 13, the behaviors of not putting on a mask and adhering to safe physical distancing measure were framed to problematize (and dissuade) such behaviors. Furthermore, by drawing emphatic similarities comparisons to suicide, it suggests that the consequence of such inaction (i.e., not abiding to COVID-19 public health measures) is equivalent to courting a suicidal death due to the increased chances of contracting COVID-19.

Our current findings appear to be aligned with deficient coping as one of the four sources of psychological strains. As the COVID-19 pandemic is a global crisis, many lives and livelihoods are naturally affected, and individuals may thus struggle to cope. This was highlighted by Topic 19: Pleas for financial aid to save lives; Topic 25: Increased suicide risks of military veterans; and Topic 20: Negative affect during COVID-19 pandemic. While it is not known if there are other sources of psychological strain experienced by individuals who died by suicide, the current study cannot rule out the effects of the remaining three psychological strains: conflicting social values, aspiration and reality incongruence, and relative deprivation [8,9]. Individuals could be triggered by conflicting social values that are demanded of them during current pandemic crisis in the form of one’s dual roles as a doctor and a mother. Working full shifts and extra overtime hours due to manpower shortage versus being unable to be home to shoulder the role and responsibilities of being a supportive mother demonstrate the conflicting social values and roles. Incongruence between aspiration and reality can take the form of a recent university graduate who would have found a job three months after graduation under normal circumstances but is currently still unemployed and unable to find a job. This was supported by Topic 36: High rates of COVID-19 mortality, unemployment, and suicide; and Topic 19: Pleas for financial aid to save lives. Lastly, on the possibility of relative deprivation, studies have showed that COVID-19 pandemic had a lesser adverse impact on some individuals than others and exposed certain socioeconomic inequalities [45,46].

Similar discussion themes, such as “public health measures to slow the spread of COVID-19” and topics related to “lockdown”, COVID-19 “deaths”, “mental health and COVID-19”, and “protests against the lockdown” were found in previous studies that analyzed text from the Twitter and Reddit social platforms [47,48,49]. Additionally, other supporting themes include “COVID-19-related death”, “economic impact”, and “preventive measures” [50]. However, discussion themes—such as global COVID-19 cases beyond China, the Diamond Princess cruise ship, discussion on supply chain [50], stigma related to COVID-19, and COVID-19 in the US [47]—were notably absent in the present study. While the present study and the two studies by Xue et al.’s [47,50] share parallel designs, the different time period for data collection and the contrasting search terms could explain the differences in findings. The data collection period for Xue et al.’s two studies [47,50] were relatively shorter, from January 2020 to March 2020 and March 2020 to April 2020, and their search terms were COVID-19-related hashtags, while the current data collection period was over a one-year period and used search terms specific to COVID-19 and suicide. Hence, salient topics trending in specific months may not be detected when there are other more prevailing topics from other time periods (e.g., Theme 5: Reported cases and public reactions to news related to COVID-19, suicide, and homicide).

Moreover, during the current data collection period, human movement restrictions with varied severity were already in place in several countries to curb the spread of COVID-19 [51,52]. Individuals may experience psychological fatigue and depressive and suicidal thoughts as a result of the lockdowns [53,54]. This may explain the more salient themes identified in present research that gravitate towards COVID-19 public health measures, biopsychosocial impact of the COVID-19 pandemic, and mental health support for at-risk individuals (i.e., Themes 1, 2, and 4, respectively).

Taken together, the six themes show how suicide and COVID-19 were publicly discussed during the COVID-19 pandemic. The general public was not only concerned about governments’ responses to curb the spread of COVID-19, there were also concerns about the biopsychosocial impact the pandemic had on mental health, suicide, economy, and at-risk populations. How the COVID-19 pandemic impacted other causes of death was also a key interest, as Twitter users compared mortality rates of COVID-19 against suicide and other causes of death. In addition, reported cases and public reaction to news related to COVID-19, suicide and homicide, and figurative usage of the word suicide were part of the larger Twitter discourse. Research should continue to examine the sociological and psychological consequences of the COVID-19 pandemic and its profound impact on persons with mental health issues, vulnerable populations, and the general public.

Nonetheless, there are several study limitations to be mentioned. This study is limited to Tweets in the English language and on the Twitter platform. The methods were unable to distinguish between tweets that reflect a discussion of suicide versus suicidal ideation. Moreover, as suicide and mental health may still be taboo topics and heavily stigmatized subjects, it is worth investigating the discourse of suicide with individuals who deemed them as such and whether the universal distress brought on by the COVID-19 pandemic has changed the way suicide and mental health are discussed in the public sphere. The public discourse surrounding suicide could also be further analyzed in relation to lifestyle, behavioral, and environmental factors. Lastly, while the findings may still reflect public sentiment related to suicide during the COVID-19 pandemic, it is possible that Twitter may have a negativity bias [55] that may influence our findings and interpretations.

## 5. Conclusions

Using unsupervised machine learning and thematic analysis techniques, the current study examined public discourse on suicide and COVID-19 during the pandemic. The general public was generally concerned about governments’ responses as well as the disquieting collateral effects on mental health, suicide, the economy, and at-risk populations. Particularly during crisis periods when policymakers require near real-time feedback to guide policy, practitioners and health authorities may apply these approaches to analyze social media data to gain a better and updated understanding of what the general public is concerned with and how key subjects are discussed. 

## Figures and Tables

**Figure 1 ijerph-19-13834-f001:**
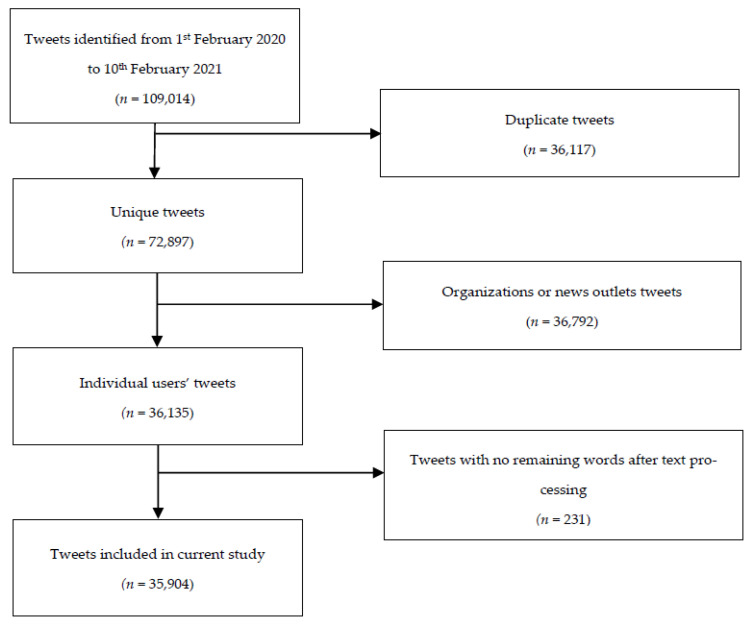
Flowchart illustrating the tweet selection process.

**Figure 2 ijerph-19-13834-f002:**
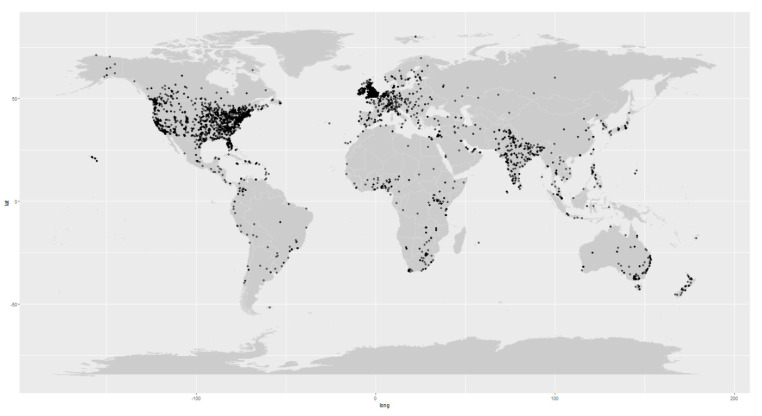
Geographical locations of the tweets included in this study. Each tweet is indicated by a black dot in the map.

**Figure 3 ijerph-19-13834-f003:**
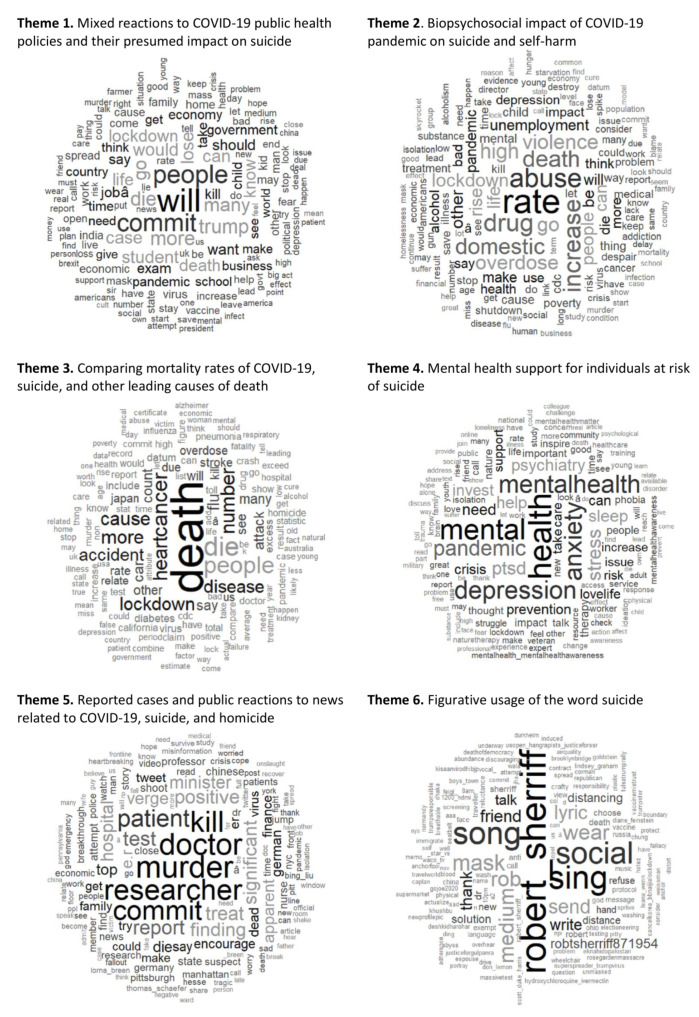
Word clouds for the six themes related to COVID-19 and suicide; the word clouds have been weighted using term frequency–inverse document frequency (TF-IDF) to give more emphasis to words that are unique to each tweet.

**Table 1 ijerph-19-13834-t001:** The six themes on Twitter related to suicide, with its respective topic labels, sample tweets, and prevalence.

Theme and Topic (*Keywords*)	Sample Tweets	Prevalence (%)
**Theme 1: Mixed reactions to COVID-19 public health policies and their presumed impact on suicide**		
Topic 41: Reactions to extended lockdowns (*will, come, end, lead, big, leave, bring, normal, wave, agree*)	“BIG sh!t coming in South Africa! BIG. SH!T. Covid19inSA LOCKSOUTHAFRICADOWN CoronavirusInSA CoronavirusPandemic Lockdown extended!!!? Economic Suicide!!!”	3.1
Topic 4: Skepticisms about the governments’ COVID-19 pandemic management (*would, should, fear, man, spread, put, right, point, quarantine, infect*)	“WAIT! WAIT! PENCE believes, and WELCOMES the RAPTURE. Putting him in charge of the US response to the coronavirus is putting a SUICIDAL PSYCHOTIC in charge of the weapons depot.”	3.0
Topic 3: Criticisms about many more deaths from lockdowns than from COVID-19 pandemic (*more, many, know, fact, wonder, bet, people, death, mask, die*)	“Did you know that deaths are up in many areas as a result of the lockdowns? Did you know there were many more drug overdose deaths in San Francisco alone than COVID-19 deaths. Drug overdoses, suicides, delayed medical care, etc. Many more people died as a result of the lockdowns”	2.9
Topic 9: Inculpating the politicians for job losses and suicide death tolls (*lose, job, economy, business, loss, pay, farmer, money, cost, fail*)	“45 million jobs lost No COVID-19 plan 174,000 Americans dead Russian bounties on US soldiers Economic crash of 33% Farmers ruined, committing suicide at 2× the national average Record debt, deficits, & trade deficits It’s your fault, Republicans”	2.5
Topic 29: Students’ protests over taking exams during COVID-19 pandemic (*case, student, give, exam, india, situation, govt, force, sir, cancel*)	“We are unable to write exams in this pandemic situation. The students are not ready to write exams in this COVID-19 situation. Please cancel all exams. WakeUpMHRD cancelfinalyearexams cancelallexams NoExamMHRD NoExamsInCovid ExamsInCovidASuicide”	2.5
Topic 30: Criticisms about President Trump holding political rallies during COVID-19 pandemic (*trump, mass, political, lie, america, president, hold, leader, american, vote*)	“James Warren was an American cult leader who conspired with his inner circle to direct a mass murder-suicide using poisoned koolaid Donald John Trump is an American cult leader who conspired with his inner circle to direct a mass murder using the coronavirus I see no difference”	2.3
Topic 26: Criticisms about governments’ response to COVID-19 pandemic (*government, home, uk, stay, deal, policy, protect, act, damage, safe*)	“Second outbreak of COVID-19 confirmed in China and we are opening the shops while we still have hundreds of death in the UK. I don’t care what this suicidal government has to say. I’m staying at home and not seeing anyone. Stay safe, people!”	2.3
Topic 18: Protests against governments’ COVID-19 response (*want, country, world, live, day, year, understand, allow, last, whole*)	“Students are not a testing kit. COVID-19 cases are increasing day by day and government is thinking to experiment on the lifes on final year students. Is this justified? ExamsInCovidASuicide”	2.1
Topic 17: What matters during COVID-19 pandemic (*real, effect, ask, plan, prevent, only, matter, global, listen, epidemic*)	“Minister Actionsto appoint a Deputy CMO in mental health gives real meaning to his commitment that ““All lives matter”“—Preventing COVID19-related suicide epidemic is now as important as preventing any other deaths! Oz now leads world on this!”	2.0
Topic 5: Advocating for reopening of schools during COVID-19 lockdowns (*child, school, kid, open, shut, teen, push, parent, small, reopen*)	“but no one cares about kids increased suicide unless they have COVID. Please please please open the schools. I’m a pediatrician drowning in kids with depression and anxiety because of your school closures”	1.6
Topic 6: Accountability for the consequences of COVID-19 lockdown measures (*may, measure, restriction, such, ignore, few, consequence, responsible, huge, possible*)	“One of Fesi’s witnesses is testifying that the COVID-19 restrictions contributed to her son’s suicide. I have no doubt that the lockdown contributed to his mental illness and that of millions of other people. That is awful. lalege lagov”	1.4
**Theme 2: Biopsychosocial impact of COVID-19 pandemic on suicide and self-harm**		
Topic 20: Negative affect during COVID-19 pandemic (*go, think, make, be, can, let, do, way, stop, thing*)	“If I was going to commit suicide right now it’s not because I’m depressed because I can’t go out. It’s because the covid19 situation is hopeless because there’s no cure or vaccine and a bunch of morons won’t wear masks and are going to get me sick”	5.2
Topic 37: Likelihood of rise in suicide rates during the COVID-19 pandemic (*increase, rise, impact, spike, infection, link, affect, population, evidence, level*)	“Not having evidence of a rise in deaths by suicide is better than having evidence of a rise of course. But, given the expected rise due to economic impacts of COVID19, what would be reassuring is specific supports for those losing jobs & income. SuicidePrevention ExcludedUK”	2.6
Topic 32: Suicide and other medical illnesses are overshadowed by COVID-19 pandemic (*other, bad, problem, illness, seem, reason, condition, long, human, cure*)	“It isn’t only Coronavirus that is killing people. From newspaper reports there seems to be a huge rise in suspected suicides. If people don’t have money, that causes all sorts of other health problems. People are also not getting help for other serious conditions”	2.5
Topic 12: Associating substance misuse and suicide with COVID-19 lockdowns (*drug, overdose, cdc, age, consider, americans, shutdown, alcohol, despair, accord*)	“75,000 Americans at risk of dying from overdose or suicide due to coronavirus despair, group warns”	2.0
Topic 36: High rates of COVID-19 mortality, unemployment, and suicide (*rate, high, unemployment, continue, low, blame, survival, skyrocket, surpass, death*)	“ALSO, SUPER high unemployment rates, suicide rates, COVID-19 rates, and racism is also at an all time high. But SURE, Donald, let’s focus on the ReAt JoBs NuMbErS.”	1.9
Topic 19: Pleas for financial aid to save lives (*life, save, face, suffer, destroy, financial, create, question, enough, answer*)	“Dear Ma’am, I’m from India. From few months begging help from everyone in twitter but no answer. Due to COVID-19 my furniture business has been destroyed completely. Facing INR 15Lacs Rupees of financial crisis. Thinking to do suicide now. Can u plz help Myfamily with any amount??”	1.8
Topic 1: Missed diagnosis and treatment delay for other medical conditions (*treatment, medical, poverty, lack, miss, delay, starvation, hunger, common, like*)	“Memo to morons. Covid19 classified as common cold in all medical journals since 1984 and right at the bottom of the list of highly infectious diseases. Missed cancer diagnosis and treatment increased suicide from lockdown induced depression thinking through your arse Morgan.”	1.5
Topic 35: Associating the exacerbation of domestic abuse, drug abuse, and gun violence to COVID-19 pandemic (*abuse, use, domestic, violence, gun, addiction, substance, alcoholism, divorce, child*)	“The COVID-19 lockdown has caused divorces, family estrangement, domestic abuse, child sexual abuse denial because no one to report outside of the abusing family. Suicide count is up as is alcoholism, drug use, over-eating, depression. Human spirit is diminished.”	1.3
Topic 42: Negative impact of COVID-19 lockdown measures on young people (*work, young, must, start, much, place, most, mention, can, make*)	“A trade-off that must be reconciled with with [sic] suppressing coronavirus is the mental health cost, especially suicide and especially young people. The economy must be restarted but we must avoid another lockdown.”	1.3
Topic 31: Victims of traumatic events (*believe, victim, research, police, turn, black, suspect, father, son, scientist*)	“Violent riots, murder & suicide, family tragedy (4 dead: mother, father, doughter [sic] and son who killed them and also killed himself), attack on the police officer—all in one week. That’s Slovenia during more than half a year of Covid19 restrictions. You sleep well?”	0.8
**Theme 3: Comparing mortality rates of COVID-19, suicide, and other leading causes of death**		
Topic 7: Excess mortality during COVID-19 pandemic (*death, count, relate, cause, result, include, excess, list, stat, non*)	“No excess deaths since June. No excess respiratory deaths, no excess coronavirus deaths. Deaths at home have accelerated above the five year average. Incidents of suicide are increasing every week. Utter disgrace.”	4.6
Topic 11: Highest death toll amongst various leading causes of death (*people, die, old, likely, failure, mask, death, more, believe, wear*)	“10 February 2020 has been the worst day in so far history of Coronavirus as 108 people lost their lives in China. But on the same day 26,283 people died of cancer. 24,641 people died of Heart disease. 4300 people died of Diabetes. More than 1000 people died of suicide.”	3.9
Topic 16: Comparing mortality rates of COVID-19 with suicide (*number, due, us, show, datum, toll, compare, total, figure, add*)	“In Japan, govt statistics show suicide claimed more lives in October than COVID-19 has over the entire year to date. The monthly number of Japanese suicides rose to 2153 in October, according to Japan’s National Police Agency. As of Friday, Japan’s total COVID-19 toll was 2087”	2.9
Topic 28: Mortality rates of COVID-19 against other leading causes of death (*cancer, disease, heart, flu, accident, attack, car, stroke, homicide, diabetes*)	“U.S. COVID19 COVID 285,564 Heart disease: 655,381 Cancer: 599,274 Accidents (unintentional injuries): 167,127 Chronic lower respiratory diseases: 159,486 Stroke: 147,810 Alzheimer’s disease: 122,019 Diabetes: 84,946 Influenza and Pneumonia: 59,120 Intentional self-harm: 48,344”	2.7
Topic 10: Positive effect of lockdown on suicide rates in Japan (*lockdown, cause, japan, woman, hit, factor, crime, surge, harm, concern*)	“Very interesting and hopeful piece... Japan suicides decline as COVID-19 lockdown causes shift in stress factors”	2.6
Topic 2: Claims of a year worth of suicide seen by a California doctor (*say, see, have, lock, california, true, australia, claim, wrong, worth*)	“A California doctor said his hospital had seen “a year’s worth of suicide attempts” in four weeks because of the coronavirus lockdowns. It was picked up by conservative media outlets to bolster comments made by Trump. But the doctor’s claim wasn’t true.”	2.6
**Theme 4: Mental health support for individuals at risk of suicide**		
Topic 38: Mental health support for suicide at-risk loved ones (*need, can, take, help, look, support, talk, one, friend, feel*)	“Yes loneliness big problem but you can SaveLives Take SuicidePrevention online training & check people are ok & get help they need. Almost 1.2 million completions of the training now. COVID19 ZeroSuicide LonelyLoss EndLoneliness”	4.5
Topic 23: Mental health crisis experienced by healthcare workers during COVID-19 pandemic (*health, mental, care, issue, crisis, public, worker, healthcare, concern, physical*)	“RT Health care worker suicides hint at COVID-19 mental health crisis to come We must care for out frontline workers”	3.0
Topic 25: Increased suicide risks of military veterans (*pandemic, risk, new, study, expert, veteran, adult, canada, ideation, experience*)	“Military and veteran suicides have increased over 20% during the pandemic. Failing to associate COVID-19 and military suicide increases is a failure in leadership.”	2.7
Topic 8: National response to suicide prevention during COVID-19 pandemic (*prevention, great, important, response, national, service, part, community, youth, action*)	“Thanks to [sic] for his continued focus on lifting our national and local responses to suicide prevention; in the Covid19 era moving urgently to lift our efforts is a national health priority”	2.4
Topic 27: Encouraging quotes for mental health awareness (*ptsd, anxiety, depression, stress, sleep, psychiatry, invest, mentalhealth, lovelife, nature*)	“Every flower is a soul blossoming in nature.~ Gerard De Nerval depression anxiety SuicidePrevention SuicideAwareness PTSD invest mentalhealth MentalHealthAwareness COVID19 lovelife stress sleep AddBrain_Inc Psychiatry nature naturetherapy”	1.9
Topic 22: Mental health awareness and suicide prevention during COVID-19 pandemic (*depression, anxiety, thought, trauma, symptom, loneliness, disorder, severe, difference, brain*)	“Help for coping with anxiety, depression, loneliness, and suicidal thoughts due to the Covid19 pandemic. Latest article for mentalhealth MentalHealthAwareness”	1.3
Topic 24: Advocating for better mental health support (*mentalhealth, good, stress, change, control, mind, little, power, happy, death*)	“mentalhealth is a global issue. On SuicidePrevention day lets try and do better and talk more about mentalhealth in these difficult times of COVID-19. Lets talk more to the people we work with and the people we love. There is no health without good mentalhealth beatNTDs”	1.2
**Theme 5: Reported cases and public reactions to news related to COVID-19, suicide, and homicide**		
Topic 40: Combatting the spread of COVID-19 and self-harm misinformation and disinformation (*report, hope, thank, read, watch, story, post, fight, article, tweet*)	“The volume of tweets and Yoda like syntax, indicate this account is probably a bot. Reported several tweets with misleading charts linking masks to COVID-19 spikes for encouraging self-harm. I can’t believe Twitter still doesn’t have a COVID-19 misinformation category.”	2.6
Topic 21: Death of a finance minister over the economic state from COVID-19 (*commit, economic, state, become, minister, crisis, worry, break, worried, track*)	“German finance minister Thomas Schaefer, 54, has committed suicide after becoming deeply worried over how to cope with the economic fallout from coronavirus”	2.5
Topic 14: Suicide death of an emergency room doctor (*virus, get, could, doctor, treat, top, fall, line, system, remember*)	“‘She had described to him an onslaught of patients who were dying before they could even be taken out of ambulances.’ COVID-19 coronavirus Top E.R. Doctor Who Treated Virus Patients Dies by Suicide”	2.4
Topic 15: Cases of suicide death and attempted suicide during COVID-19 pandemic (*family, test, patient, hospital, positive, try, attempt, ppl, member, tragic*)	“A Malayali nurse who attempted suicide after testing positive for coronavirus passed away in Haryana. Bismi Scaria, a native of Punalur, was a nurse at a private hospital in Haryana’s Gurugram. She attempted suicide on May 28 after the disease worsened by hanging herself.”	2.2
Topic 39: News reports of the death of a COVID-19 researcher who was close to making significant findings (*kill, murder, dead, close, researcher, significant, finding, chinese, apparent, shoot*)	“Chinese researcher on the verge of making ‘very significant’ Coronavirus findings is killed in murder-suicide”	2.0
Topic 34: News reports related to COVID-19 and suicide (*time, isolation, news, mean, follow, run, drive, nation, food, area*)	“In this roundup: news that suicide in Japan has dropped by 20% during COVID-19 because people are spending more time with family, less time commuting, and less time in school.”	1.9
**Theme 6: Figurative usage of the word suicide**		
Topic 33: Metaphorizing COVID-19 vaccine as suicide (*call, find, tell, vaccine, person, hear, sad, china, dr, well*)	“…The coronavirus vaccine is the ‘final solution’ depopulation weapon against humanity; globalists hope to convince BILLIONS of people to commit ‘suicide-via-vaccine’.”	2.4
Topic 13: Metaphorizing refusal for mask wearing and social distancing as suicide (*social, mask, medium, wear, hand, send, distancing, refuse, contact, anti*)	“Not wearing a face mask and social distancing, is committing suicide by COVID-19.”	1.2

## Data Availability

The data supporting the reported results are available from the corresponding author on reasonable request.

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
