# Peer review of "Public Discourse Surrounding Suicide during the COVID-19 Pandemic: An Unsupervised Machine Learning Analysis of Twitter Posts over a One-Year Period"

_ijerph, 2022, doi:10.3390/ijerph192113834_

Round 1
Reviewer 1 Report
This study thus aimed to examine how individuals relate to suicide 14
amid the COVID-19 pandemic from a global perspective viz the public Twitter discourse around suicide and COVID-19.
I really liked reading this manuscript. I found the topic interesting and methodology and data presentation impressive. It is in the spirit of improving the manuscript that I provide the following questions and/or recommendations:
In Your Abstract you wrote:
Abstract: Many studies have forewarned the profound emotional and psychosocial impact of the protracted COVID-19 pandemic. This study thus aimed to examine how individuals relate to suicide amid the COVID-19 pandemic from a global perspective viz the public Twitter discourse around suicide and COVID-19. Original Twitter tweets from 1 February 2020 to 10 February 2021 were searched, with terms related to ‘COVID-19’, ‘suicide’, or ‘self-harm’. An unsupervised machine-learning approach and topic modelling were used to identify topics from unique tweets, with each topic further grouped into themes using manually conducted thematic analysis by the study investigators. A total of 35,904 tweets related to suicide and COVID-19 were processed into 42 topics and 20
six themes. The main themes were: (1) mixed reactions to COVID-19 public health policies; (2) impact of the pandemic on suicide and self-harm; (3) comparing mortality rates of COVID-19, suicide, and other leading causes of death; (4) mental health support for individuals at risk of suicide; (5) public reactions to news related to COVID-19, suicide, and homicide; and (6) figurative usage of the word suicide. The general public was generally concerned about governments’ responses as well as the perturbing effects on mental health, suicide, the economy, and at-risk populations.
Change to:
In Your Abstract you wrote:
Abstract: Many studies have forewarned the profound emotional and psychosocial impact of the protracted COVID-19 pandemic. This study thus aimed to examine how individuals relate to suicide amid the COVID-19 pandemic from a global perspective via the public Twitter discourse around suicide and COVID-19. Original Twitter tweets from 1 February 2020 to 10 February 2021 were searched, with terms related to ‘COVID-19’, ‘suicide’, or ‘self-harm’. An unsupervised machine-learning approach and topic modelling were used to identify topics from unique tweets, with each topic further grouped into themes using manually conducted thematic analysis by the study investigators. A total of 35,904 tweets related to suicide and COVID-19 were processed into 42 topics and 20
six themes. The main themes were: (1) mixed reactions to COVID-19 public health policies; (2) impact of the pandemic on suicide and self-harm; (3) comparing mortality rates of COVID-19, suicide, and other leading causes of death; (4) mental health support for individuals at risk of suicide; (5) public reactions to news related to COVID-19, suicide, and homicide; and (6) figurative usage of the word suicide. The general public was generally concerned about governments’ responses as well as the perturbing effects on mental health, suicide, the economy, and at-risk populations.
On Page 3, you wrote:
Majority of tweets with location information were from North America (25.9%) and 119 Europe (11.5%), with remaining from Asia (5.2%), Australia (2.6%), Africa (1.6%), South 120 America (0.6%), and unknown location (52.7%).
Change to:
The majority of tweets with location information were from North America (25.9%) and 119 Europe (11.5%), with remaining from Asia (5.2%), Australia (2.6%), Africa (1.6%), South 120 America (0.6%), and unknown location (52.7%).
On Page 17, you wrote:
Nonetheless, there are several study limitations to be mentioned. This study is limited to Tweets in English language and on Twitter platform.
Change to:
Nonetheless, there are several study limitations to be mentioned. This study is limited to Tweets in the English language and on the Twitter platform.
OTHER ISSUES:
· The page numbers are not correct.
· At the time that this manuscript was reviewed, the number of individuals that have become infected or died from COVID-19 have diminished. What, if anything, do you believe Twitter users will say about this decline? Does society have an inclination to only focus on the negative and ignore the positive?

Author Response
Thank you for the kind suggestions.
- We have made the edits to the text as suggested by the reviewer. Changes in the manuscript were highlighted in yellow.
- We have corrected the page numbers. Apologies for this.
- Thank you for the comment. This is difficult to conjecture as most Asian countries still see COVID-19 as a real and present threat. Although the death rates have declined especially with greater availability of vaccines and therapeutics, COVID-19 remains a global pandemic. Many countries are seeing new surges because of the Omicron subvariants and COVID still has the propensity to cause severe illnesses with a higher death rate. Nonetheless, we have now added in our discussion of study limitations that "Twitter may have a negativity bias that may influence our findings and interpretations."
Reviewer 2 Report
This study examines the discourse around suicide and the COVID-19 pandemic. Given the public concern around the potential impact of the global pandemic on suicide, this systematic examination of the discourse on the topic makes it a valuable contribution to the body of literature on the impact of the COVID-19. As highlighted by the author, it also has the potential to provide a template for gathering near real time feedback on prevailing sentiment in times of a public crisis. I do, however, have some minor shortcomings.
The introduction is well-written overall.
Line 42-44. Are the figures ranges? In other words, do the predictive models predict an increase in suicide deaths of between 2,135 and 9,570 per year and job losses of between 5.3 and 24.7 million? I recommend rewording this sentence to improve clarity.
Methods
It would be helpful if you could provide some detail about the publicly available COVID 19 twitter dataset. Also, indicate why you picked that particular timeframe.
It would be helpful to document which of the study investigators conducted thematic analysis, and whether each investigated independently coded all data or if the dataset was divided among the investigators with some element of cross-checking?
Reading the results, each tweet seems to have been assigned to one of the themes only. In the methods, it would be helpful to outline that this was the approach taken and why, as I imagine some tweets could have come under more than one theme if they made multiple points.
Results
For theme 6, the name Robert Sherriff is the most common word in the word cloud, but it is not mentioned in table 1, or in the main text. Some context on why this word is so prominent here (but not a key word in the two topics that comprise this theme) would be helpful.
Discussion
Overall, I recommend refining the discussion considering the following points.
Line 198-200 – It is not clear how this point relates to the points being made in paragraph – it would be helpful if this was elaborated upon.
In the paragraph on theme 4 from line 232 to 238, I recommend elaborating on the specific social concern you mention, as “the importance of mental health” is vague. It is ambiguous what the source of this statement on line 235 is – make it clear it refers to individuals rallying others on twitter (if that is correct).
In the discussion on theme 5, is there evidence to support saying that the twitter discourse was about COVID, homicide and suicide (248-250)? This is just the focus of this theme, and the dataset was specifically COVID-related so I am not sure what point is being made here.
Also in relation to theme 5, given the nature of the study and the focus on public discourse, the authors should be cautious to not over-estimate the conclusions that can be drawn from their findings in relation to causes of suicidal behaviour.
Paragraph from lines 292 to 296 does not make a very clear point – it could be made better by removing the specific quotes referenced and referring to similarities with other studies in a more reflective way.
Minor comments
Opening sentence written in present tense – past tense would read better.
Page 1 line 34 – recommended replacing the word ‘multitudes’
Row 193, remove word “but”
Lines 205-207- I recommend re-writing this sentence to improve readability, there are a number of prepositions missing
Author Response
- Apologies for the ambiguity. Yes the figures are ranges. We have corrected the sentence, which now reads "Predictive models estimated an increase of approximately 2,135 to 9,570 suicides per year, in association to projected figures of 5.3 to 24.7 million ..."
- Thank you for the comments. We have now added that "... publicly available COVID-19 Twitter dataset, which contained chatter related to COVID-19 between 1 January 2020 and 27 Jun 2021 [20]." As public sentiment would change over time, we chose the specific time period to reflect more updated findings as compared to earlier works (which utilized social media data from January 2020 to March 2020 and March 2020 to April 2020) on the topic.
- Thank you for the comments. We have further elaborated on our methods, "Output from topic modelling was examined by three study investigators (S.R.L., Q.X.N. and T.M.L.) to ensure coherence of the identified topics. Descriptive labels for the topics were manually crafted by the same study investigators based on identified keywords and sample tweets of each topic. Thereafter, the topics were further grouped into themes by the two authors (TML and CSL) Lastly, thematic analysis was performed through an inductive and iterative process as introduced by Braun and Clarke (2006), with (1) study investigators (S.R.L., Q.X.N., X.X., Y.L.L., E.S.K.B., and T.M.L) familiarizing themselves with the keywords and sample tweets; (2) producing preliminary codes; (3) formulating overarching themes; (4) reviewing and refining themes; (5) defining and specifying themes; and (6) producing a write-up [22]."
- Thank you for the comment. We have now specified which of the study investigators were involved in performing the thematic analysis, and that each investigated independently coded all data.
- Thank you for the comment. We have relooked at the data and with regard to the name "Robert Sherriff", "The name ‘Robert Sherriff’ was the most common word in the word cloud, who may have multiple Twitter accounts (poet, singer, author, historian etc.). The other words “sing”, “song” and “robertsheriff871954” could have been referring to his COVID-19 song, which expressed pessimism that, “COVID-19 has got me crying … I got so many friends out there just dying.” Notably, some of these words in the word cloud (Figure 3) were not identified in Table 1. Reasons for the discrepancy between Table 1 keywords and Theme 6 word cloud could be that "Term Frequency–Inverse Document Frequency (TF-IDF), which was used to generate the word clouds, may give more emphasis to words that are unique to each theme and larger items in the word cloud could indicate more informative (higher TF-IDF scoring) features [23]." We have explained this in our results section.
- We have refined the discussion section, incorporating the suggestions made by the reviewer. "Theme 4: Mental health support for individuals at risk of suicide during COVID-19 pandemic highlighted a pertinent social concern, where many patients faced limited access to psychiatric services and social isolation during the pandemic [35]. Topic 24: Advocating for better mental health support highlighted individuals rallying others on twitter during such a trying period."
- We have tempered our discussion regarding Theme 5: "Several high-profile COVID-19 related suicide and homicide cases in the news entered the Twitter public discourse, forming the larger proportion of topics within Theme 5, e.g. “German minister commits suicide after ‘coronavirus crisis worries’.” While it is premature to draw conclusions on the causes of suicidal behaviour during COVID-19 based on news reports, this seems to be aligned with STS deficient coping as one of the apparent sources of psychological strains experienced by individuals who reportedly died by suicide [8]."
- The opening sentence is now written in past tense as suggested.
- The word 'multitudes' was replaced with 'multiple facets'.
- The word 'but' was removed from line 193 as suggested.
- Rewrote sentence in lines 205-207, with missing propositions now added.
Reviewer 3 Report
Comments for authors attached as separate file

Author Response
Thank you for the useful comments and references! Our replies below and changes in the manuscript were highlighted in yellow for easy identification.
- Thank you for the comment. STS provides a theoretical lens to conceptualize and interpret the potential adverse effects of the COVID-19 pandemic on suicidal thoughts and behaviour, and the current findings. As for our methodology, we used an inductive approach where the themes surfaced based on original coding through machine learning, hence the coding structure was not directly influenced by the STS framework.
- Thank you for the comment. Unfortunately, given the large volume of tweets and inherent constraints of the dataset, we were unable to distinguish between tweets that reflect a discussion of suicide versus suicidal ideation. We have added this in our discussion of study limitations.
- Thank you for the comments. We have further elaborated on our methods, "Output from topic modelling was examined by three study investigators (S.R.L., Q.X.N. and T.M.L.) to ensure coherence of the identified topics. Descriptive labels for the topics were manually crafted by the same study investigators based on identified keywords and sample tweets of each topic. Lastly, thematic analysis was performed through an inductive and iterative process as introduced by Braun and Clarke (2006), with (1) study investigators (S.R.L., Q.X.N., X.X., Y.L.L., E.S.K.B., and T.M.L) familiarizing themselves with the keywords and sample tweets; (2) producing preliminary codes; (3) formulating overarching themes; (4) reviewing and refining themes; (5) defining and specifying themes; and (6) producing a write-up [22]. Inter-coder reliability was not calculated as the authors reviewed the codes and themes independently, coding disagreements between authors were resolved by further discussion until consensus was reached."
- The word 'topic' was used in place of 'code' as we used machine learning techniques, specifically topic modelling technique, which was applied to identify clusters of words that tend to co-occur. High probability co-occurring words are considered to be of same ‘topic’ as per previous studies [17]. Topic modelling allows a large number of texts to be analysed using machine-learning algorithms, thereby reducing the amount of time taken to analyze data manually.
- In our paper, we used thematic analysis as described by Braun and Clarke (2006) as this is an accessible and theoretically flexible approach to analysing qualitative data. This was thought to be a more suitable complement to machine learning approaches as it can be applied to complex data and across a range of theoretical and epistemological approaches. We have further elaborated on this in our methods section.
- Thank you for the comment. As aforementioned, STS was a framework that provides a theoretical lens to conceptualize and interpret the potential adverse effects of the COVID-19 pandemic on suicidal thoughts and behaviour, and the current findings. As for our methodology, we used an inductive and iterative approach where the themes surfaced based on original coding through machine learning.
Round 2
Reviewer 3 Report
The authors did a good job with revisions.